

# Children with abnormal body weight dealing with the load of school backpack—is there a need to modify WHO recommendations? A cross sectional study

Anna Brzęk, Regina Wysocka-Bochenek and Jacek Sołtys

Department of Physiotherapy, Faculty of Health Sciences in Katowice, Medical University of Silesia in Katowice, Katowice, Poland

Corresponding author
Anna Brzęk, abrzek@sum.edu.pl

## ABSTRACT

**Background**. The purpose of the study was investigated how stabilographic parameters change under the influence of an external load, such as a school backpack, and whether there is a need to revise recommendations regarding the permissible school backpack load for children with abnormal body mass.

**Methods**. Children in younger school grades was investigated (n, 235, age: $X = 7.90 \pm 0.74$), divided into children with underweight (n = 49), with overweight (n = 48), and with obesity (n = 33), in comparison to the control group with normal body weight (n = 105). Posturographic measurements of body weight distribution and posturometric tests with eyes open and closed were performed in trials with school backpack and without.

**Results**. The weight of school backpack for younger children ranges from 1.50 to 8.0 kg ($3.74 \pm 1.31$). On average, this constitutes 13.66% of their body weight (SD, 5.85), but in 77 cases among all participants (32.77%), the children's school backpacks were overloaded by an average of 5.5% (SD, 4.66). Neither the weight of the school backpack nor its percentage relative to body weight were predictors of postural stability ($-0.44 \geq \beta \geq 0.41$ for $p > 0.05$). In tests involving a school backpack, children with obesity performed significantly worse compared to the control group, particularly in the parameters sway path length (SPL), width of the ellipse (WoE), and height of the ellipse (HoE), in both open- and closed-eye conditions. Additionally, SPL was longer in the open-eye test with a school backpack in the overweight group. Children with underweight had results comparable to children with normal body weight.

**Conclusions**. Recommendations regarding the weight of school backpacks relative to a child's body weight worsen stabilometric parameters in children with increased body weight (overweight and obesity), which affects their postural stability. It is advisable to consider different percentage load values or implement systemic solutions for children with overweight and obesity.

## INTRODUCTION

The issue of overloaded school backpacks remains unresolved. For many years, researchers have been attempting to assess various aspects such as the curvature of the spine under unilateral load (*Drzał-Grabiec et al., 2015*), the impact of time spent walking with a backpack on posture changes (*Kistner et al., 2013*), and comparing carrying a backpack with a double-sided bag (differences in children when carrying a traditional backpack *versus* a double-sided one) (*Mosaad & Abdel-Aziem, 2018*). Numerous studies also address the normative weight values for backpacks (*Dockrell, Blake & Simms, 2015*; *Al-Saleem et al., 2016*; *Spiteri et al., 2017*; *Kasović, Štefan & Zvonar, 2019*). These studies recommend a load ranging from 10% to 25% of the child's body weight, depending on the region or country. However, there is no global consensus on the weight limit for backpacks, with the most commonly cited limit being up to 15%. The authors point to the negative effects of carrying a load, such as forward body tilt (*Bruno et al., 2012*; *Drzał-Grabiec et al., 2015*), pain (*Dockrell, Simms & Blake, 2016*), and spine issues (*Dockrell, Simms & Blake, 2015*), although in the latter case, there are many confounding factors that may influence the occurrence of these issues, and the studies are observational (*Kistner et al., 2013*; *Talbott et al., 2009*). Research on school backpacks is important, as the immediate effects of carrying overloaded backpacks (*Barbosa et al., 2019*) are well known, such as the occurrence of muscle pain (*Negrini & Carabalona, 2002*; *Sheir-Neiss et al., 2003*; *Lindstrom-Hazel, 2009*; *Dockrell, Simms & Blake, 2015*; *Calvo-Muñoz et al., 2020*; *Hernández et al., 2020*), including lower back pain (LBP), or discomfort associated with wearing them (*Dockrell, Blake & Simms, 2015*). However, it is difficult to determine the long-term effects due to the lack of studies.

Few studies evaluate stabilometric parameters under the load of a school backpack, which seem to be one of the key factors in postural stability disturbances. Postural stability control is an integral part of every motor task, with particular emphasis on complex tasks, and its peak development occurs at a younger age (*Tan, 2019*). Postural control depends on expanding the limits of stability, influencing strategies (*Shumway-Cook & Woollacott, 1985*; *Colné et al., 2008*). However, how the additional weight generated by a school backpack affects stabilometric parameters in children remains unanswered in the literature. Research in this area is lacking. Moreover, none of the existing studies on backpack weight include children with abnormal body weight. When analyzing children with underweight, overweight, and obesity, who are subject to the same backpack weight standards, we cannot be certain whether the recommended loads are appropriate. Overweight and obese children, according to current recommendations, are allowed to carry a heavier school backpack, which, given their skeletal system's burden with excess body mass, is already harmful even without the additional weight of a backpack. Underweight children, on the other hand, must carry a lighter backpack despite having the same lessons on a given day, which may make it difficult to meet this criterion, exposing underweight children to overload of the musculoskeletal system (*Jeon & Kim, 2021*; *Brooks et al., 2021*).

The authors of this study, based on empirical research, clearly state that when selecting the weight of a school backpack, a broader perspective should be taken to assess how the

load is distributed across the support base and whether the additional weight on the child's back affects postural stability, which plays a key role in developing proper body posture.

Based on the gap in the evidence-based literature, the aim of this study was to examine the average actual percentage ratio of the backpack weight to the body weight of children with abnormal body mass in relation to the recommended normative values. Additionally, the study investigated how stabilographic parameters change under the influence of an external load, such as a school backpack, and whether there is a need to revise recommendations regarding the permissible backpack load for children with abnormal body mass.

## THE MATERIALS AND METHODS

### Ethics statement

This study was a cross-sectional observational study. The children from Silesian Poland Schools were screened. Eligible children were invited to participate in the study. All parents/guardians gave their written informed consent prior the subjects' enrollment in the study. The study was conducted in accordance with the guidelines of the Declaration of Helsinki reviewed and approved by The Bioethics Committee of Medical University of Silesia in Katowice, Poland (Decision no. KNW/022/KB1/100/I/16/18 and BNW/NWN/0052/KB1/124/II/22/23/24).

### Study participant

The sample size was calculated using G*Power 3.1 software, assuming a significance level of $\alpha = 0.05$, 95% test power, and a medium effect size of 0.25, indicating a representative group of 280 subjects. A loss and data incompleteness rate of 20% was accounted for during the main study (assessment of stabilometric parameters). An invitation to participate in the study was sent to 1,600 randomly selected younger school-aged children. The parents of the children were invited to participate in a study conducted in primary schools in the Silesian Voivodeship during parent–teacher meetings held in public schools. The invitation was also sent to parents *via* the student electronic information system in the respective schools. The groups were similar at baseline regarding the most important prognostic indicator as a age. Six-year-old children ($n = 87$), who had started school a year earlier, were immediately excluded from the study. The remaining parents/legal guardians were asked to complete a consent form for the study. A total of 987 complete consents were received, of which an additional 20% were excluded due to the absence of signatures from both parents in legally regulated situations. We used a proprietary survey questionnaire designed for the initial screening of invited children (a written interview). It was completed by parents and included questions regarding, among other things, the child's year of birth, coexisting diseases, treatment in specialist clinics, injuries sustained within the past six months, use of orthoses, and physical activity. Further children were excluded from the study due to exclusion criteria, the most important of which were: lack of a completed questionnaire by the parents, lower limb injuries within 3 months before the start of the study, current musculoskeletal injuries, significant hearing and vision impairments, lower limb shortening, foot orthotics, diagnosed postural defects with particular emphasis on the sagittal plane, and serious spinal deformities such as scoliosis, Scheuermann's disease, or

motion sickness, heart defects, those who had undergone heart defect surgery in infancy, or those with pulmonary issues such as asthma. The study included children wearing a school backpack on both shoulders. Children carrying a school backpack on one shoulder or holding it in one hand were excluded from this study. The study included children who met the WHO criteria for physical activity at a moderate-to-vigorous level. They participated in physical education classes at school. No child was physically inactive, nor did they engage in professional sports training. The group was homogeneous in this regard. Information about the level of physical activity was obtained from the questionnaire. In the next stage of qualification—assessing the children's posture, an additional 34% of the children were excluded due to identified postural abnormalities, injuries, infections, and students from sports-oriented classes. Ultimately, 294 children aged 7–9 years ($8.39 \pm 1.12$) were qualified for the main study—assessment of stabilometric parameters. From the final analysis of results, 13% of the data were excluded due to artifacts or incomplete records in certain trials (this value falls within the 20% data loss threshold initially assumed in the study). Additionally, during the course of the study, five participants withdrew, and 10% had an incompletely loaded school backpack on the day of the examination. The statistical analysis was conducted on 235 children ($X = 7.91 \pm 0.74$), among whom there were 128 girls and 107 boys. The participants were divided into two groups based on body mass: Group I—children with abnormal body mass (n = 130), and Group II—children with normal/healthly body mass (n = 105). The criterion for group classification was based on percentile charts for the Polish population (Ola and Olaf, PL) (*Kułaga et al., 2023*). BMI was calculated by dividing weight in kilograms by the square of height in meters (kg/m2) based on the WHO BMI chart using the BMI $z$-score criteria, which is underweight, normal weight, overweight, obese, and severely obese (*DeOnis & Lobstein, 2010*). The BMI $z$-score was calculated using the WHO AnthroPlus (*World Health Organization, 2024*) software reference date for 5–19 years, with an indication of healthly with values $1 \geq z$ score BMI $\geq$ -2, overweight > +1.0 standard deviation (SD), obese > +2.0 SD (an extended cutoff from the WHO growth curve) (WHO, 2007). The body weight was controlled by tared medical scale to the nearest 0,1 kg and body height was measured using a growth meter to the nearest 0.1 cm (Tanita DC-430MA). To both measurements the children were required to stand barefoot and without clothes. The first group was further divided into three subgroups: Ia—underweight children (n = 49), Ib—overweight children (n = 48) and Ic—obese children (n = 33). The characteristics of the study and II—control groups are presented in Table 1.

## Methods

### The weight of the school backpack evaluation

The weight of the school backpack (SW) was assessed using a tared medical scale. Then, the % ratio of school backpack weight to body weight was calculated in accordance with the recommendations of the General Sanitary Inspectorate (GIS), taking 10–15% of body weight as normative values (*Chief Sanitary Inspectorate (GIS) of Poland, 2024*) A division was made into normative backpacks (N) and overloaded backpacks exceeding the recommended value (ON). Weighing of the school bags was done in the morning with a

**Table 1   Characteristics of the groups.**

| Parameters | Ia (*n* = 49) | Ib (*n* = 48) | Ic (*n* = 33) | II (*n* = 105) | K-W | *Post hoc*[1] |
|---|---|---|---|---|---|---|
| | X ± SD (Range) | X ± SD (Range) | X ± SD (Range) | X ± SD (Range) | | |
| Age (years) | 7.98 ± 0.76 (7–9) | 7.98 ± 0.79 (7–9) | 7.79 ± 0.77 (7–9) | 7.85 ± 0.71 (7–9) | 2.02 | – |
| Height (*z*-score)* | 0.6 ± 1.02 (−1.91–3.09) | 0.93 ± 0.96 (−0.67–3.33) | 1.29 ± 1.13 (−1.33–3.32) | 0.48 ± 0.92 (−1.61–3.05) | 23.34 | <0.01 {Ia/Ib; Ib/II} <0.001 {Ia/Ic; Ic/II} |
| Body weight (*z*-sore)* | −1.21 ± 0.83 (−4.0–0.31) | 1.51 ± 0.48 (0.78–2.54) | 2.62 ± 1.07 (−1.63–4.86) | 0.28 ± 0.61 (−1.15–2.09) | 188.15 | <0.0001 {Ia/Ic; Ia/Ib; Ia/II; Ib/II; Ic/II} |
| BMI (*z*-score)* | −2.55 ± 0.58 (−4.18–−2.02) | 1.39 ± 0.27 (1.01–1.98) | 2.72 ± 0.64 (2.13–4.71) | 0.01 ± 0.49 (−1.24–0.98) | 195.68 | <0.0001 {Ia/Ic; Ia/Ib; Ia/II; Ib/II; Ic/II} |
| Sex, n (%) | | | | | | |
| Girls | 31 (60.78) | 32 (62.75) | 13 (44.83) | 52 (50.0) | – | – |
| Boys | 20 (39.22) | 19 (37.250 | 16 (55.17) | 52 (50.0) | | |

**Notes.**

Abbreviations: Ia, group with underweight; Ib, group with overweight; Ic, group with obese; II, group with healthly weight; X, average; SD, standard deviation; K-W, Kruskal–Wallis test.

[1] ANOVA Tukey'a post-hoc.

*For Ola and Olaf recommendation.

full bag load. School backpacks were weighed throughout the entire week, but since the timetable schedule for children in grades I–III did not differ significantly between days, the backpack weight was also not statistically significant between measurements ($p > 0.05$).

### *Posturographic test with open and closed eyes*

The methodology applied in another study by the authors was also used in this study. The research was conducted using the Zebris PDM platform (Isny, Germany) in a standing position, without socks, with the arms relaxed alongside the body, and the subject's gaze directed forward and focused on a single point at nose level. Tests were conducted with open eyes (Open-Eye) and closed eyes (Closed-Eye). Two trials were carried out: one without a backpack and one with a backpack (Fig. 1) symmetrically worn on both of the subject's shoulders. Each measurement consisted of three trials, each lasting 30 s, with a 30-second break between trials. The average values of the results from these three trials were used for analysis.

The study was conducted at the school to avoid unpacking school backpacks. The research conditions were ensured: the study took place in the school nurse's office. Silence was maintained during the examination. When planning the study, the class schedule was taken into account; the examinations were conducted only during lessons. The school bell schedule was also considered to eliminate the risk of disturbances such as noise.

The following posturographic parameters were analyzed: %BWD$_{(P/L)}$ (*Body Weight Distribution*) between the right and left sides of the body (*DeBlasiis et al., 2023*; *Bukowska et al., 2021*) and COP (center of pressure)-related parameters: MCoCX—mean of points with $x$-coordinates (medial-lateral direction), MCoCY—mean of points with $y$-coordinates (posterior-anterior direction), SPL—sway path length, WoE—width of the ellipse, HoE—height of the ellipse, AoE—area of the ellipse, aoE—angle of the ellipse. For the analysis of

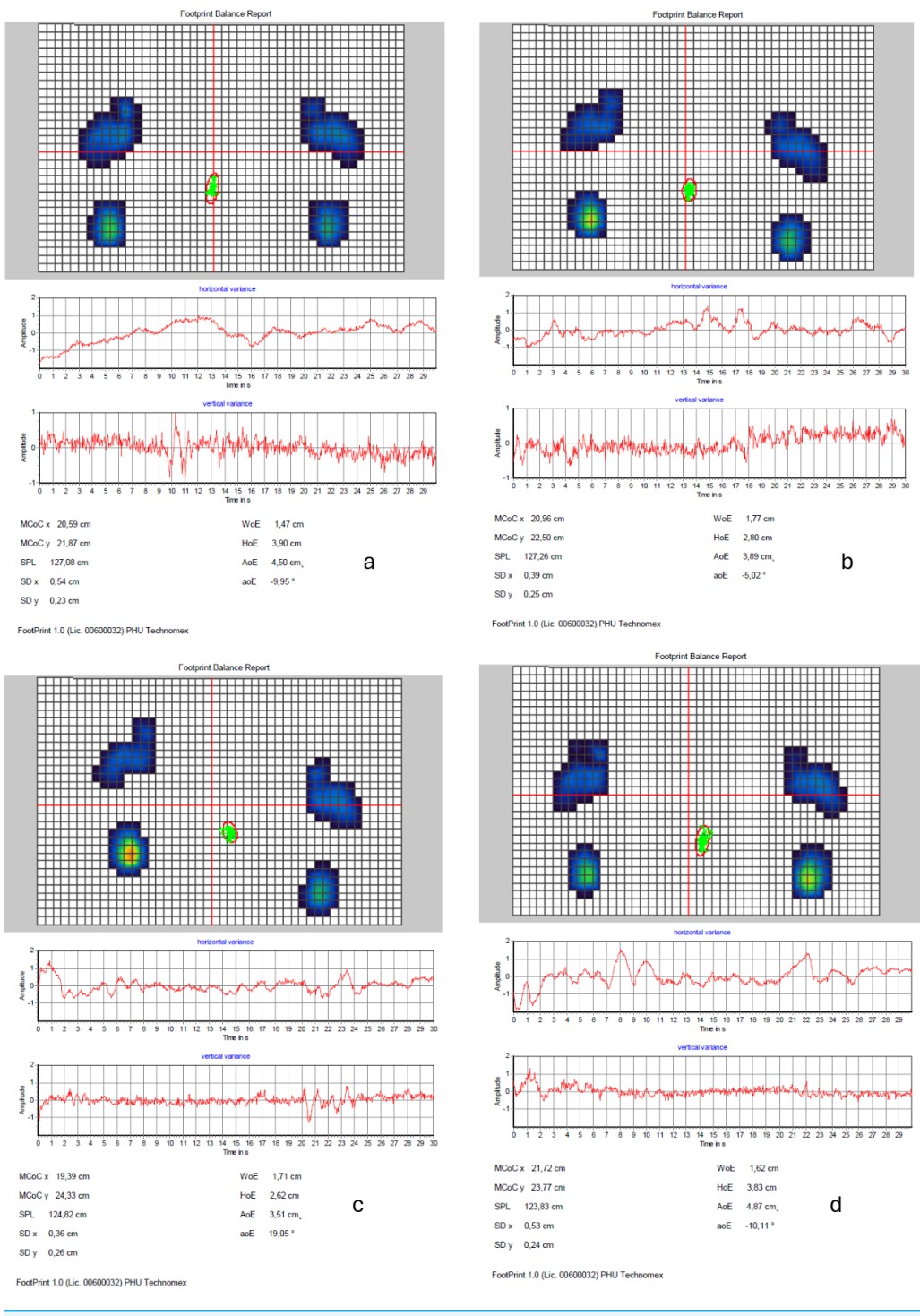

**Figure 1 Zebris PDM.** (A) Without a backpack; eyes open; (B) without a backpack; eyes closed; (C) with a backpack; eyes open; (D) with a backpack; eyes closed. A graphical representation of an example foot loading distribution (Footprint Balance Report), sway amplitude in horizontal and vertical variance, and the results of stabilographic parameters.

stabilometric parameters were assessed. The load distribution (BWD) indicates either the symmetry or asymmetry of the load between the right and left sides. COP parameters—the position of the center of pressure of the feet under static conditions—graphically represent the process of standing position control. SPL indicates the length of the trajectory of the body's overall center of gravity during the test. The data analysis considers the average deviation of foot pressure in the sagittal plane in the antero-posterior direction (relative to the $y$-axis) and in the frontal plane in the medio-lateral direction (relative to the $x$-axis). Additionally, the outermost points of COP displacement, forming the envelope of an ellipse, are analyzed, with the size (width, height), shape and area, and angle of inclination of the ellipse being evaluated.

### Statistical analysis

Statistical analyses were carried out using the Excel and Statistics 13v programs. Descriptive statistics of the studied parameters—medians (Me), means (X), standard deviations (SD). Non-parametric statistics were used for intergroup comparisons: the Man–Whitney $U$ test for two groups and Kruskal–Wallis ANOVA—when there were more groups. Multiple comparisons were performed using the Tukey's $p$-value. Missing data analysis and collinearity diagnostics were conducted before the analyses. The assumed level of statistical significance: $p < 0.05$. Homogeneity of variance was verified with the $F$-test, F-Snedecor test. The Shapiro-Wilk test was used to evaluate the normal distribution. To assess changes in stabilometric parameters, the Student's $t$-test and the Wilcoxon signed-rank test were used. Linear regression models were then used to explore how each of the predictors were associated with the stabilometric parameters. The correlation was calculated using Pearson's R and R-Spearmann when the distribution was not considered to be normal.

## RESULTS

First, the results related to the weight of the school backpack are presented, followed by the analysis of stabilometric parameters.

### School backpacks weight

The weight of school backpacks among younger grade children ranges from 1.50 to 8.0 kg ($3.74 \pm 1.31$). Typically, this constitutes 13.66% of their body weight (SD, 5.85), but in 77 cases among the total participants (32.77%), the school backpacks are overloaded by an average of 5.5% (SD, 4.66). However, it should be noted that in the group of underweight children, this overload exceeding recommendations affects as many as 63% of the respondents, compared to only 10% among overweight children and 0.09% among obese children. For children with normal body weight, this value exceeded one-third (Table 2).

No differences were observed between groups in the actual weight of the school backpacks; however, as expected, differences were noted in the amount (in kg) exceeding the recommended norms ($p < 0.0001$).
**Table 2  Schoolbags' parameters in the groups taking age division into account.**

| Parameters | Ia (n = 49) | Ib (n = 48) | Ic (n = 33) | II (n = 105) | K-W | Post hoc[1] |
|---|---|---|---|---|---|---|
| | X ± SD (Range) | X ± SD (Range) | X ± SD (Range) | X ± SD (Range) | | |
| Weight (kg) | 3.78 ± 1.39 (1.60–7.60) | 3.64 ± 1.09 (1.60–5.50) | 3.92 ± 1.28 (1.50–6.60) | 3.74 ± 1.38 (1.60–8.00) | 1.22 | – |
| % SW vs. BW | 17.14 ± 7.02 (7.14–38.19) | 11.15 ± 3.45 (5.31–19.57) | 10.19 ± 3.96 (3.52–20.96) | 14.27 ± 5.61 (6.20–28.57) | 35.31 | <0.001 {Ia/Ib; Ib/Ic; Ib/II; Ic/II} |
| % ↑ GIS norm | 5.94 ± 6.02 (0.17–23.19) | 2.27 ± 1.94 (0.11–4.57) | 3.82 ± 2.53 (1.02–5.96) | 5.51 ± 3.57 (0.19–13.57) | 3.99 | – |

Notes.

Abbreviations: Ia, group with underweight; Ib, group with overweight; Ic, group with obesity; II, group with healthly weight; %SW vs. BW, % schoolbags weight vs. body weight %; ↑ GIS norm, % above the GIS norm (Polish abbreviation - Chief Sanitary Inspectorate); X, average; SD, standard deviation; K-W, Kruskal–Wallis test.

[1]ANOVA Tukey'a post-hoc.

The weight of the backpack for all girls was comparable to the weight of the backpacks for boys (3.77 vs. 3.69; $p = 0.62$). Gender did not determine backpack weight in any group (all $p > 0.15$).

## Posturographic parameters
### Assessment of body weight distribution between the left and right side during an Eyes-Open and Eyes-Closed test

The difference in weight distribution between the right and left side ($\Delta$BWD) in tests with and without a backpack during the Eyes-Open and Eyes-Closed trials is shown in Fig. 2.

No significant difference between the groups in individual trials was observed ($p = 0.317$). An increase in the asymmetry difference between sides was noted in the trial without a backpack during the Eyes-Closed test compared to the Eyes-Open test in groups Ia, Ib, and II, as well as a decrease in $\Delta$BWD in group Ic, but these were not statistically significant ($p = 0.46$, $p = 0.247$, $p = 0.55$, and $p = 0.16$, respectively). In the test with a backpack, results in both trials were comparable (all $p > 0.23$). In the underweight and overweight groups, a phenomenon of asymmetry equalization in foot loading was noted ($X^2$Ia = 8.26, $df = 3$, $p = 0.04$ and $X^2$Ib = 16.06, $df = 3$, $p < 0.001$, respectively), though it significantly worsened other stabilometric parameters (Tables 3 and 4).

### Assessment of balance parameters during tests with the Eyes Open and Closed

The studied groups were homogeneous in terms of the distribution of stabilometric parameters and did not differ from each other in the open-eyes test without a backpack or in the closed-eyes test (for all parameters, $p > 0.06$), except for the ellipse angle, which decreased after closing the eyes (Table 3).

However, the situation was markedly different in the test with a backpack. Children with obesity performed significantly worse compared to the control group, particularly in the parameters SPL, WoE, and HoE, both in the open-eyes and closed-eyes tests. Additionally, SPL was longer in the open-eyes test with a backpack in the overweight group. Children with underweight had results comparable to those with normal body weight (Table 4).

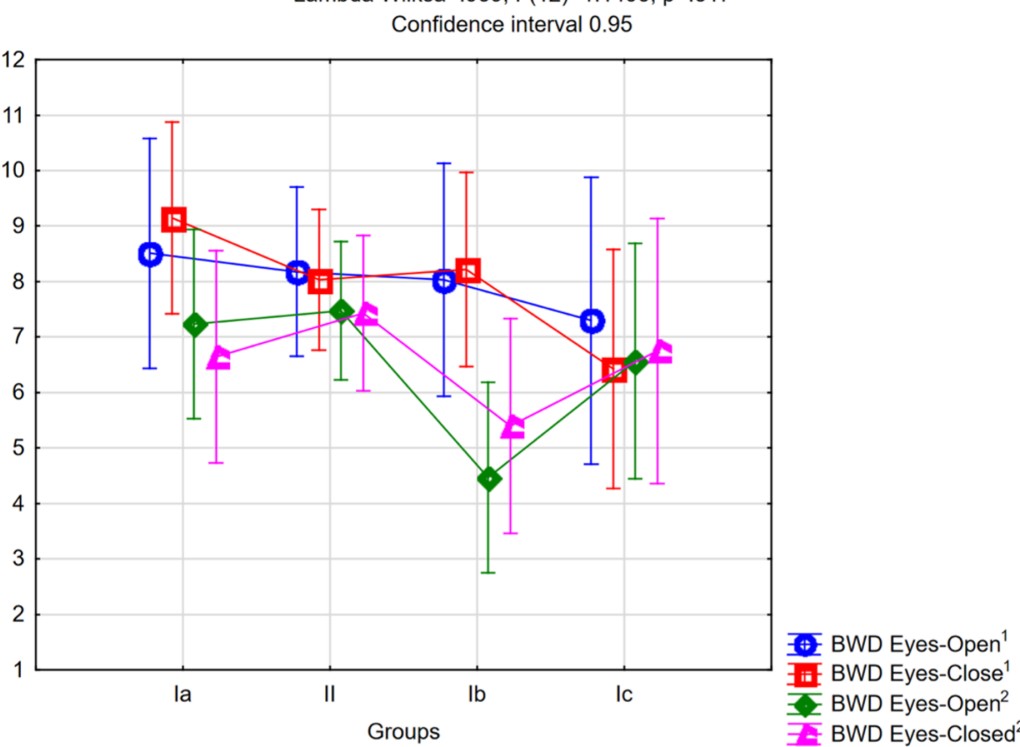

**Figure 2** **The difference in weight distribution (BWD) between the right and left side in tests with and without a backpack during the Eyes-Open (1) and Eyes-Closed trials (2) in fall measurement group.** BWD Eyes Open1—body weight distribution during Eyes-Open trials in test without a backpack; BWD Eyes Close1—body weight distribution during Eyes-Close trials in test without a backpack; BWD Eyes Open2—body weight distribution during Eyes-Open trials in test with a backpack; BWD Eyes Close2—body weight distribution during Eyes-Close trials in test with a backpack.

Attention should be paid to changes in parameter results when closing the eyes. In many cases, this indicates the use of proprioception mechanisms in postural control (BWD in group IIb). Significant differences were noted in the parameters SPL, WoE, HoE, and AoE. In tests without a backpack, closing the eyes worsened results across all groups, highlighting the critical role of visual control in maintaining a stable body position. Similar results were observed in tests with a backpack, except in group Ic (with obesity), which utilized the proprioception mechanism. The additional weight of the school backpack significantly increased the center of pressure path length (SPL) in children with overweight (Ib) and obesity (Ic) when their eyes were open, a phenomenon not observed in children with underweight or normal body weight. In all children, wearing a backpack led to a broadening of the ellipse, and in the group of children with obesity, it also caused an increase in its height (Table 5).

### The predictors and their correlations with posturometric parameters

In the following section, predictors that may influence stabilometric outcomes were identified, namely: BMI (percentiles), school backpack weight (SW), and the percentage of

Brzęk et al. (2025), *PeerJ*, DOI 10.7717/peerj.19353

**Table 3 Comparison of the results of posturographic tests conducted during free-standing position without a schoolbag between the groups (Ia,Ib,Ic,II) in trials: Eyes Open and Eyes Closed.**

| Posturographic parameters | Eyes - Open | | | | Eyes - Closed | | | | ANOVA | |
|---|---|---|---|---|---|---|---|---|---|---|
| | Ia | Ib | Ic | II | Ia | Ib | Ic | II | Tukey's p-value | |
| | M (SD) | M (SD) | M (SD) | M (SD) | M (SD) | M (SD) | M (SD) | M (SD) | $p_1$ | $p_2$ |
| ∆BWD | 8.28 (6.44) | 8.10 (6.79) | 7.87 (6.78) | 7.98 (8.86) | 8.86 (4.47) | 8.43 (6.28) | 6.94 (6.27) | 8.07 (6.57) | 0.993 | 0.556 |
| MCoCx | 8.21 (7.23) | 10.18 (8.14) | 16.25 (34.44) | 10.31 (10.11) | 10.09 (8.49) | 11.25 (10.31) | 10.11 (8.27) | 10.89 (9.88) | 0.131 | 0.911 |
| MCoCy | 14.12 (8.51) | 13.24 (9.53) | 16.10 (15.03) | 15.92 (12.14) | 14.52 (10.73) | 12.98 (11.93) | 15.52 (16.42) | 14.47 (11.29) | 0.496 | 0.818 |
| SPL (mm) | 228.06 (189.17) | 244.26 (119.99) | 224.76 (87.36) | 254.19 (129.07) | 292.14 (134.87) | 317.84 (123.83) | 298.45 (132.21) | 309.53 (146.38) | 0.451 | 0.718 |
| WoE (mm) | 7.23 (3.19) | 9.09 (6.03) | 7.74 (4.13) | 8.66 (5.37) | 8.84 (7.64) | 9.71 (5.28) | 6.84 (3.85) | 9.42 (6.72) | 0.227 | 0.179 |
| HoE (mm) | 11.65 (5.05) | 14.23 (10.40) | 13.74 (7.76) | 14.01 (8.34) | 13.85 (6.52) | 17.07 (9.05) | 14.34 (6.95) | 15.88 (10.26) | 0.361 | 0.276 |
| AoE ($mm^2$) | 181.17 (118.85) | 345.79 (589.44) | 211.91 (168.95) | 289.87 (357.69) | 284.74 (310.22) | 381.41 (344.38) | 235.88 (241.38) | 360.33 (651.51) | 0.117 | 0.474 |
| aoE (°) | 15.00 (17.00) | 23.21 (22.37) | 16.06 (16.81) | 23.95 (22.72) | 14.93 (17.01) | 17.91 (19.41) | 13.34 (14.04) | 16.69 (16.58) | <0.05[ab] | 0.621 |

**Notes.**

Abbreviations: Ia, group with underweight; Ib, group with overweight; Ic, group with obesity; II, group with healthly weight; M, Mean Values; SD, Standard Deviation; ∆BWD$_{(P/L)}$, Body Weight Distribution between right and left side; MCoCX, Mean of points with $x$-coordinates (medial-lateral direction); MCoCY, Mean of points with $y$-coordinates (posterior-anterior direction); SPL, COP path length; (III), the COP area; WoE, width of the ellipse; HoE, height of the ellipse; AoE, area of the ellipse; aoE, angle of the ellipse.

$p_1$-value statistical significance tests results in the Eyes-Open trial; $p_2$-value statistical significance tests results in the Eyes-Closed trial. All statistical analysis were conducted using the ANOVA with using the Tukey's $p$-value: [a] difference between Ia and Ib; [b] difference between Ia and II.

**Table 4  Comparison of the results of posturographic tests conducted during free-standing position with a schoolbag between the groups (Ia,Ib,Ic,II) in trials: Eyes Open and Eyes Closed.**

| Posturographic parameters | Eyes - Open | | | | Eyes - Closed | | | | ANOVA Tukey's *p*-value | |
|---|---|---|---|---|---|---|---|---|---|---|
| | Ia M (SD) | Ib M (SD) | Ic M (SD) | II M (SD) | Ia M (SD) | Ib M (SD) | Ic M (SD) | II M (SD) | p1 | p2 |
| ΔBWD | 6.88 (5.84) | 4.54 (4.04) | 7.10 (5.43) | 7.45 (6.73) | 6.51 (5.78) | 5.37 (4.21) | 7.33 (9.51) | 7.65 (6.81) | 0.02[abcd] | 0.05[d] |
| MCoCx | 8.35 (7.59) | 7.31 (5.48) | 10.77 (9.95) | 9.93 (9.46) | 7.06 (5.31) | 7.12 (7.11) | 10.52 (12.91) | 9.81 (8.94) | 0.191 | 0.091 |
| MCoCy | 16.77 (14.91) | 14.06 (5.48) | 20.27 (13.55) | 15.56 (11.44) | 16.96 (15.37) | 14.81 (14.92) | 19.03 (17.14) | 14.07 (10.47) | 0.163 | 0.255 |
| SPL (mm) | 276.40 (151.47) | 295.49 (173.37) | 447.19 (68.50) | 272.23 (127.51) | 372.87(118.01) | 341.12 (179.04) | 427.31 (94.05) | 319.75 (137.39) | <0.0001[def] | <0.001[bf] |
| WoE (mm) | 11.29 (12.51) | 11.27 (7.10) | 15.59 (8.60) | 10.66 (6.65) | 11.12 (4.74) | 11.09 (4.07) | 14.07 (6.75) | 10.54 (5.43) | 0.036[cf] | 0.011[f] |
| HoE (mm) | 12.78 (6.91) | 16.39 (12.49) | 18.67 (7.52) | 14.65 (7.98) | 15.57 (8.38) | 17.99 (10.39) | 21.47 (11.26) | 16.68 (8.28) | 0.021[ef] | 0.029[ef] |
| AoE (mm$^2$) | 397.26 (160.40) | 466.61 (624.68) | 372.56 (439.95) | 359.72 (353.45) | 372.44 (403.01) | 467.24 (499.33) | 426.39 (398.81) | 404.02 (355.24) | 0.743 | 0.697 |
| aoE (°) | 24.71 (20.24) | 30.61 (27.54) | 25.30 (25.06) | 24.03 (20.43) | 17.95 (17.59) | 17.21 (20.86) | 18.38 (17.82) | 21.47 (20.01) | 0.406 | 0.541 |

**Notes.**

Abbreviations: Ia, group with underweight; Ib, group with overweight; Ic, group with obesity; II, group with healthly weight; M, Mean values; ΔBWD(P/L), Body weight distribution between right and left side; MCoCX, Mean of points with *x*-coordinates (medial-lateral direction); MCoCY, Mean of points with *y*-coordinates (posterior-anterior direction); SPL, COP path length; (III), the COP area; WoE, width of the ellipse; HoE, height of the ellipse; AoE, area of the ellipse; aoE, angle of the ellipse. p1-value statistical significance test results in the Eyes-Open trial; p2-value statistical significance test results in the Eyes-Closed trial. All statistical analysis were conducted using the ANOVA with using the Tukey's *p*-value: [a] difference between Ia and Ib; [b] difference between Ia and II; [c] difference between Ib and Ic; [d] difference between Ib and II; [e] difference between Ia and Ic; [f] difference between Ic and II.

school backpack weight relative to the subject's body weight (%SW). Stepwise regression analysis was used to identify potential predictors for specific variables in the posturographic tests presented in the table below (Table 6) for tests with a backpack under both open-eye and closed-eye conditions.

Table 6 summarizes descriptive statistics and the results of parameter analysis for posturographic measurements with eyes open and closed. As indicated by the regression results, in the tests with eyes open, only the variables McoCy and HoE correlate with the school backpack weight. In tests with a backpack under closed-eye conditions, only WoE shows a correlation. Subsequently, it was examined whether the backpack weight and the proportion of backpack weight relative to body weight (Table 7) are related to posturographic parameters.

It turned out that this relationship is present only in part and primarily in the underweight group and the group with normal body weight. In the open-eye tests, in the underweight group, there is a linear relationship between school backpack weight and SPL, WoE, HoE, AoE, and aoE, as well as for the percentage of backpack weight relative to body weight in both tests. In the group of children with normal body weight, closing the eyes amplifies the correlation of the aforementioned parameters with the school backpack weight, but this relationship is not observed when analyzing the percentage load. Detailed correlations are presented in Table 7.

## DISCUSSION

The aim of this project was to evaluate the weight of school backpacks among younger school-age children and determine whether the child's body mass is related to postural stability deficits. Since the normative weight of backpacks is calculated as a percentage of the child's body mass, it raised the question of whether the recommended values should be the same for children with abnormal body mass. There is a lack of scientific evidence

**Table 5  Comparison of the results of posturographic tests conducted during free-standing position between tests without a schoolbag and with a schoolbag in trials: Eyes Open and Eyes Closed.**

| Posturographic parameters | Tests | Ia | Ib | Ic | II |
|---|---|---|---|---|---|
| | | Z; p | Z; p | Z; p | Z; p |
| ΔBWD | Eyes-Open[1]/Eyes-Closed[1] | 0.73; 0.461 | 0.72; 0.470 | 1.39; 0.163 | 0.59; 0.554 |
| | Eyes-Open[2]/Eyes-Closed[2] | 0.14; 0.885 | 1.32; 0.184 | 0.54; 0.583 | 0.01; 0.998 |
| | Eyes-Open[1]/Eyes-Open[2] | 0.87; 0.381 | **2.81; 0.005** | 0.69; 0.486 | 0.09; 0.921 |
| | Eyes-Closed[1]/Eyes- Closed[2] | 1.59; 0.112 | **2.087; 0.036** | 0.53; 0.596 | 0.68; 0.492 |
| MCoCx | Eyes-Open[1]/Eyes-Closed[1] | 0.87; 0.381 | 1.01; 0.312 | 0.17; 0.859 | 0.19; 0.845 |
| | Eyes-Open[2]/Eyes-Closed[2] | 0.29; 0.770 | 0.87; 0.381 | 0.69; 0.486 | 0.19; 0.845 |
| | Eyes-Open[1]/Eyes-Open[2] | 0.14; 0.885 | 1.16; 0.243 | 0.10; 0.99 | 0.01; 0.997 |
| | Eyes-Closed[1]/Eyes- Closed[2] | 1.58; 0.112 | **2.33; 0.02** | 0.18; 0.859 | 0.88; 0.378 |
| MCoCy | Eyes-Open[1]/Eyes-Closed[1] | 0.41; 0.663 | 0.43; 0.665 | 0.34; 0.727 | 1,58; 0.115 |
| | Eyes-Open[2]/Eyes-Closed[2] | 0.43; 0.665 | 0.001; 0.999 | 0.17; 0.859 | 1.68; 0.092 |
| | Eyes-Open[1]/Eyes-Open[2] | 0.72; 0.471 | 0.43; 0.665 | **2.78; 0.005** | 0.88; 0.377 |
| | Eyes-Closed[1]/Eyes- Closed[2] | 0.58; 0.559 | 0.1; 0.99 | 1.23; 0.216 | 0.78; 0.435 |
| SPL (mm) | Eyes-Open[1]/Eyes-Closed[1] | **3.89; <0.0001** | **5.06; <0.0001** | **2.78; = 0.005** | **4.88; <0.00001** |
| | Eyes-Open[2]/Eyes-Closed[2] | **2.16; 0.031** | **2.62; 0.008** | **4.17; <0.00001** | **3.71; 0.0002** |
| | Eyes-Open[1]/Eyes-Open[2] | 1.29; 0.193 | **3.608; 0.0003** | **5.22; <0.0001** | 1.76; 0.078 |
| | Eyes-Closed[1]/Eyes- Closed[2] | **3.61; 0.0003** | 0.43; 0.665 | **4.18; <0.0001** | 1.56; 0.118 |
| WoE (mm) | Eyes-Open[1]/Eyes-Closed[1] | 0.58; 0.559 | 1.75; 0.080 | 1.74; 0.082 | **2.73; 0.006** |
| | Eyes-Open[2]/Eyes-Closed[2] | 0.43; 0.665 | 1.91; 0.055 | 0.1; 0.99 | 0.78; 0.434 |
| | Eyes-Open[1]/Eyes-Open[2] | **3.32; 0.0009** | **2.45; 0.01** | **3.36; 0.0007** | **3.23; 0.001** |
| | Eyes-Closed[1]/Eyes- Closed[2] | **2.74; 0.006** | 1.58; 0.112 | **3.82; <0.0001** | **3.04; 0.002** |
| HoE (mm) | Eyes-Open[1]/Eyes-Closed[1] | 1.87; 0.061 | **2.45; 0.01** | 1.04; 0.296 | 1.95; $p = 0.051$ |
| | Eyes-Open[2]/Eyes-Closed[2] | 2.74; 0.006 | 1.16; 0.243 | **2.08; 0.036** | **3.15; 0.001** |
| | Eyes-Open[1]/Eyes-Open[2] | 0.14; 0.885 | 1.87; 0.061 | **2.43; 0.01** | 0.19; 0.845 |
| | Eyes-Closed[1]/Eyes- Closed[2] | **2.72; 0.006** | 0.1; 0.99 | **3.83; <0.0001** | 1.66; 0.095 |
| AoE (mm²) | Eyes-Open[1]/Eyes-Closed[1] | 1.29; 0.191 | **3.03; 0.002** | 0.69; 0.486 | 1.95; 0.051 |
| | Eyes-Open[2]/Eyes-Closed[2] | 1.29; 0.194 | 0.58; 0.559 | 1.74; 0.082 | **2.93; 0.003** |
| | Eyes-Open[1]/Eyes-Open[2] | 1.58; 0.112 | 1.58; 0.111 | 0.74; 0.082 | **2.53; 0.011** |
| | Eyes-Closed[1]/Eyes- Closed[2] | **2.45; 0.001** | 0.72; 0.471 | **3.13; 0.002** | **3.13; 0.002** |
| aoE (°) | Eyes-Open[1]/Eyes-Closed[1] | 0.72; 0.471 | 0.14; 0.885 | 0.17; 0.859 | 1.96; 0.061 |
| | Eyes-Open[2]/Eyes-Closed[2] | 1.01; 0.321 | **2.91; 0.003** | 0.35; 0.728 | 0.49; 0.623 |
| | Eyes-Open[1]/Eyes-Open[2] | 1.87; 0.061 | 1.01; 0.312 | 1.39; 0.164 | 0.20; 0.844 |
| | Eyes-Closed[1]/Eyes- Closed[2] | 0.10; 0.99 | 0.72; 0.471 | 0.88; 0.378 | **2.15; 0.032** |

**Notes.**

Abbreviations: Ia, group with underweight; Ib, group with overweight; Ic, group with obesity; II, group with healthly weight; ΔBWD$_{(P/L)}$, body weight distribution between right and left side; MCoCX, Mean of points with $x$-coordinates (medial-lateral direction); MCoCY, Mean of points with $y$-coordinates (posterior-anterior direction); SPL, COP path length; (III), the COP area; WoE, width of the ellipse; HoE, height of the ellipse; AoE, area of the ellipse; aoE, angle of the ellipse.

[1] Trials without schoolbag.

[2] Trials with schoolbag.

Statistically significant results were highlighted in bold (all statistical analysis were conducted using the Wilcoxon test).

Brzęk et al. (2025), PeerJ, DOI 10.7717/peerj.19353

**Table 6  Summary of regression models for posturographic parameters in both trials (the eyes-open and eyes-closed measurement) in tests with schoolbag.**

| Variable | Predictors | Eyes - Open | | | | | Eyes - Closed | | | | |
|---|---|---|---|---|---|---|---|---|---|---|---|
| | | B | SE | β | t | p | B | SE | β | t | p |
| %BWD(P/L) | Intercept | 3.936 | 5.61 | | 0.702 | 0.483 | 7.67 | 6.17 | | 1.24 | 0.213 |
| | SW | 0.55 | 0.87 | 0.12 | 0.63 | 0.525 | 0.69 | 0.96 | 0.13 | 0.71 | 0.474 |
| | %SW | −0.05 | 0.22 | −0.06 | −0.26 | 0.793 | −0.08 | 0.24 | −0.08 | −0.37 | 0.711 |
| | BMI perceniles | 0.009 | 0.02 | 0.06 | 0.500 | 0.617 | 0.01 | 0.02 | 0.05 | 0.53 | 0.597 |
| | Age | −0.32 | 0.66 | −0.04 | −0.48 | 0.629 | −0.79 | 0.73 | −0.09 | −1.07 | 0.284 |
| | Sex | 0.42 | 0.39 | 0.07 | 1.07 | 0.285 | 0.75 | 0.44 | 0.11 | 1.71 | 0.091 |
| | | $F_{(5;227)}=0.54$; $p = 0.74$; $R^2 = 0.01$ | | | | | $F_{(5;228)}=1.22$; $p = 0.300$; $R^2 = 0.03$ | | | | |
| MCoCx | Intercept | 12.05 | 8.90 | | 1.35 | 0.177 | 10.12 | 8.15 | | 1.24 | 0.215 |
| | SW | 1.99 | 1.24 | 0.30 | 1.61 | 0.109 | 1.92 | 1.27 | 0.28 | 1.51 | 0.132 |
| | %SW | −0.32 | 0.31 | −0.22 | −1.02 | 0.308 | −0.27 | 0.32 | −0.17 | −0.83 | 0.408 |
| | BMI perceniles | −0.005 | 0.03 | −0.01 | −0.16 | 0.866 | −0.02 | 0.03 | −0.05 | −0.48 | 0.630 |
| | Age | −0.70 | 0.95 | −0.06 | −0.74 | 0.458 | −1.12 | 0.97 | −0.09 | −1.16 | 0.247 |
| | Sex | 0.39 | 0.56 | 0.05 | 0.69 | 0.485 | 0.77 | 0.58 | 0.08 | 1.34 | 0.181 |
| | | $F_{(5;128)}=1.41$; $p = 0.22$; $R^2 = 0.03$ | | | | | $F_{(5;128)}=1.72$; $p = 0.129$; $R^2 = 0.04$ | | | | |
| MCoCy | Intercept | 21.49 | 11.83 | | 1.81 | 0.071 | −0.81 | 14.26 | | −0.06 | 0.954 |
| | SW | 4.06 | 1.75 | 0.41 | 2.31 | **0.021** | −0.82 | 1.99 | −0.07 | −0.41 | 0.680 |
| | %SW | −1.01 | 0.46 | −0.45 | −2.18 | **0.030** | 0.38 | 0.51 | 0.16 | 0.76 | 0.451 |
| | BMI perceniles | −0.07 | 0.04 | −0.21 | −2.17 | 0.058 | 0.08 | 0.05 | 0.17 | 1.53 | 0.127 |
| | Age | −0.34 | 1.29 | −0.01 | −0.26 | 0.794 | 1.22 | 1.52 | 0.07 | 0.80 | 0.424 |
| | Sex | 0.62 | 0.83 | 0.04 | 0.74 | 0.456 | 1.06 | 0.91 | 0.07 | 1.16 | 0.244 |
| | | $F_{(5;228)}=1.40$; $p = 0.22$; $R^2 = 0.03$ | | | | | $F_{(5;228)}=0.99$; $p = 0.42$; $R^2 = 0.02$ | | | | |
| SPL (cm) | Intercept | 459.9 | 135.74 | | 3.38 | <0.0001 | 333.59 | 149.16 | | 2.24 | 0.026 |
| | SW | 25.73 | 20.18 | 0.22 | 1.28 | 0.202 | −15.07 | 20.83 | −0.13 | 1.28 | 0.202 |
| | %SW | −3.75 | 5.29 | −0.14 | −0.71 | 0.479 | 5.97 | 5.30 | 0.23 | 1.12 | 0.261 |
| | BMI perceniles | 0.48 | 0.46 | 0.11 | 1.05 | 0.295 | 0.76 | 0.65 | 0.15 | 1.35 | 0.177 |
| | Age | −28.75 | 14.82 | −0.14 | −1.94 | 0.054 | −6.45 | 15.91 | −0.04 | −0.41 | 0.685 |
| | Sex | 16.46 | 9.55 | 0.11 | 1.72 | 0.086 | 4.461 | 9.45 | 0.03 | 0.46 | 0.638 |
| | | **$F_{(5;228)} = 3.67$; $p = 0.003$; $R^2 = 0.07$** | | | | | $F_{(5;228)} = 1.09$; $p = 0.36$; $R^2 = 0.02$ | | | | |

**Table 6** (*continued*)

| Variable | Predictors | Eyes - Open | | | | | Eyes - Closed | | | | |
|---|---|---|---|---|---|---|---|---|---|---|---|
| | | B | SE | β | t | p | B | SE | β | t | p |
| | *Intercept* | 24.88 | 7.29 | | 3.42 | <0.0001 | 19.04 | 4.44 | | 4.28 | <0.0001 |
| | *SW* | 0.84 | 0.89 | 0.13 | 0.94 | 0.346 | 1.24 | 0.54 | 0.30 | 2.29 | 0.023 |
| | *%SW* | −0.06 | 0.21 | −0.05 | −0.29 | 0.771 | −0.16 | 0.13 | −0.18 | −1.29 | 0.195 |
| WoE | *BMI percentiles* | −0.008 | 0.02 | −0.03 | −0.37 | 0.706 | 0.01 | 0.01 | −0.06 | −0.83 | 0.406 |
| (cm) | *Age* | **−1.90** | **0.82** | **−0.16** | **−2.30** | **0.022** | −1.19 | 0.50 | −0.17 | −2.36 | 0.019 |
| | *Sex* | 0.54 | 0.56 | 0.06 | 0.97 | 0.332 | 0.81 | 0.34 | 0.15 | 2.36 | 0.019 |
| | | F (5;228) = 1.93; *p* = 0.089; $R^2$ = 0.04 | | | | | **F (5;228) = 3.71; *p* = 0.003; $R^2$ = 0.07** | | | | |
| | *Intercept* | 26.57 | 7.59 | | 3.49 | 0.0005 | 24.69 | 8.04 | | 3.07 | 0.002 |
| | *SW* | 1.78 | 0.92 | 0.26 | 1.93 | 0.050 | 0.80 | 0.98 | 0.11 | 0.82 | 0.412 |
| | *%SW* | −0.28 | 0.22 | −0.18 | −1.28 | 0.203 | −0.13 | 0.23 | −0.08 | −0.58 | 0.562 |
| HoE | *BMI percentiles* | −0.006 | 0.02 | −0.02 | −0.29 | 0.768 | −0.005 | 0.02 | −0.02 | −0.23 | 0.818 |
| (cm) | *Age* | −1.74 | 0.86 | −0.44 | −2.02 | 0.044 | −1.02 | 0.91 | −0.08 | −1.13 | 0.261 |
| | *Sex* | 0.71 | 0.59 | 0.008 | 1.21 | 0.228 | 0.02 | 0.62 | 0.003 | 0.05 | 0.963 |
| | | F (5;228) = 1.97; *p* = 0.083; $R^2$ = 0.04 | | | | | F (5;228) = 0.38; *p* = 0.861; $R^2$ = 0.008 | | | | |
| | *Intercept* | 487.72 | 478.05 | | 1.02 | 0.309 | 320.39 | 345.25 | | 0.92 | 0.354 |
| | *SW* | −23.80 | 58.20 | −0.05 | −0.41 | 0.683 | −6.94 | 42.03 | −0.02 | −0.17 | 0.869 |
| | *%SW* | 13.82 | 13.88 | 0.14 | 0.96 | 0.321 | 8.8 | 10.02 | 0.12 | 0.86 | 0.393 |
| AoE | *BMI percentiles* | −0.26 | 1.47 | −0.01 | −0.17 | 0.858 | 0.52 | 1.06 | 0.04 | 0.49 | 0.624 |
| ($cm^2$) | *Age* | −22.41 | 54.16 | −0.03 | −0.41 | 0.679 | −3.23 | 39.12 | −0.006 | −0.008 | 0.934 |
| | *Sex* | 34.11 | 36.90 | 0.006 | 0.92 | 0.356 | 26.29 | 26.65 | 0.06 | 0.99 | 0.325 |
| | | F (5;228) = 0.83; *p* = 0.525; $R^2$ = 0.02 | | | | | F (5;228) = 0.68; *p* = 0.636; $R^2$ = 0.01 | | | | |
| | *Intercept* | 14.69 | 19.36 | | 0.76 | 0.449 | 0.98 | 16.46 | | 0.06 | 0.952 |
| | *SW* | 1.93 | 3.36 | 0.12 | 0.82 | 0.412 | 0.61 | 2.00 | 0.04 | 0.31 | 0.759 |
| | *%SW* | −0.49 | 0.56 | −0.12 | −0.89 | 0.376 | 0.06 | 0.47 | 0.02 | 0.13 | 0.898 |
| aoE | *BMI percentiles* | −0.10 | 0.06 | −0.13 | −1.72 | 0.085 | −0.008 | 0.05 | −0.01 | −0.17 | 0.866 |
| (°) | *Age* | 2.09 | 2.19 | 0.07 | 0.495 | 0.342 | 2.03 | 1.86 | 0.08 | 1.09 | 0.276 |
| | *Sex* | −0.42 | 1.49 | −0.01 | −0.28 | 0.776 | 3.13 | 1.27 | 0.16 | 2.47 | 0.014 |
| | | F (5;228) = 1.09; *p* = 0.361; $R^2$ = 0.02 | | | | | F (5;228) = 1.56; *p* = 0.173; $R^2$ = 0.03 | | | | |

**Notes.**

Abbreviations: B, beta; SE, standard error; β, differentiated beta; t, *t*-test; *p*, statistical significance; F, ANOVA test; $R^2$, adjusted $R^2$- value; MCoCX, Mean of points with *x*-coordinates (medial-lateral direction); MCoCY, Mean of points with *y*-coordinates (posterior-anterior direction); SPL, COP path length; WoE, width of the ellipse; HoE, height of the ellipse; AoE, angle of the ellipse; SW, schoolbag weight (kg); %SW, percentage value of the school backpack weight relative to body weight; % BMI, Body Mass Index (percentiles); MFT, Results of the Matthias test (sek).
Statistically significant results were highlighted in bold.

Peer J

**Table 7  Summary of correlations for posturographic parameters *vs.* schoolbags' weight and its % *versus* body mass in both trials the eyes-open and eyes-closed measurement.**

| Parameters | Schoolbags' weight | | | | | | | | % Schoolbags' weight *vs.* body mass | | | | | | | |
|---|---|---|---|---|---|---|---|---|---|---|---|---|---|---|---|---|
| | Eyes - Open | | | | Eyes - Closed | | | | Eyes - Open | | | | Eyes - Closed | | | |
| | Ia[1] | Ib[2] | Ic[2] | II[2] | Ia[1] | Ib[2] | Ic[2] | II[2] | Ia[1] | Ib[1] | Ic[1] | II[1] | Ia[1] | Ib[1] | Ic[1] | II[1] |
| ΔBWD | 0.07 | 0.09 | 0.29 | −0.02 | −0.13 | 0.20 | 0.20 | 0.007 | 0.07 | 0.05 | 0.32 | 0.06 | −0.11 | 0.19 | 0.21 | 0.06 |
| MCoCx | −0.14 | −0.06 | 0.33 | 0.15 | −0.21 | −0.12 | 0.26 | 0.15 | −0.14 | −0.03 | 0.21 | −0.001 | −0.19 | −0.10 | 0.32 | 0.06 |
| MCoCy | −0.01 | −0.008 | 0.24 | −0.09 | 0.13 | −0.12 | 0.03 | **−0.26**\* | −0.003 | −0.11 | 0.32 | −0.01 | 0.14 | 0.01 | 0.30 | −0.15 |
| SPL (mm) | **0.29**\* | 0.06 | 0.29 | 0.06 | 0.27 | 0.34 | 0.24 | **0.37**\* | **0.29**\* | 0.08 | 0.29 | 0.009 | **0.34**\* | 0.001 | 0.14 | 0.08 |
| WoE (mm) | **0.39**\* | −0.09 | −0.01 | 0.18 | 0.24 | 0.08 | 0.14 | **0.39**\* | **0.39**\* | 0.02 | 0.02 | 0.315 | **0.34**\* | −0.08 | 0.16 | 0.12 |
| HoE (mm) | **0.41**\* | 0.01 | −0.03 | 0.14 | **0.29**\* | 0.16 | −0.35 | **0.44**\* | **0.41**\* | −0.07 | 0.13 | **0.21**\* | **0.36**\* | −0.03 | **−0.39**\* | 0.11 |
| AoE (mm²) | **0.41**\* | −0.007 | −0.17 | 0.15 | **0.37**\* | 0.21 | −0.07 | **0.46**\* | **0.41**\* | −0.03 | −0.18 | 0.17 | **0.47**\* | −0.05 | −0.30 | 0.14 |
| aoE (°) | **0.20**\* | **−0.29**\* | −0.19 | 0.02 | −0.03 | 0.26 | 0.11 | 0.21 | 0.20 | −0.23 | −0.03 | 0.009 | −0.04 | −0.05 | −0.09 | 0.07 |

**Notes.**

[1] Pearson's R for variables with a normal distribution.

[2] Spearman's R for data with a non-normal distribution.

\*statistical significance $p \leq 0.05$.

Abbreviations: MCoCX, Mean of points with *x*-coordinates (medial-lateral direction); MCoCY, Mean of points with *y*-coordinates (posterior-anterior direction); SPL, COP path length; WoE, width of the ellipse; HoE, height of the ellipse; AoE, angle of the ellipse.

Statistically significant results were highlighted in bold.

in this area, which motivated the authors to address this topic and fill the gap with solid evidence.

Our study was conducted with great methodological precision regarding the children's age, exclusion criteria, and applied methodology to ensure the results were as reliable as possible and applicable to the population of children in this age group. Recommendations regarding backpack weight exist only for children aged 7–9 years; therefore, this age group was our study population to allow comparison of our results with those of other researchers. Our study revealed the issue of underestimating the normative values of a school backpack in children with overweight and obesity, as well as the problem of an overloaded school backpack regardless of proper packing in underweight children. The results clearly demonstrated the significant changes in postural stability observed in children suffering from obesity. Finally, we were not unable to definitively identify predictors related to postural stability, including school backpack's weight. This was only partially successful in children underweight and normal body weight. This area requires further research with particular attention to children with obesity.

However, there is a lack of data on the burden of school backpacks in children with abnormal body weight. This issue has been highlighted by researchers from Spain, who also considered overweight and obese children (*Orantes-Gonzalez & Heredia-Jimenez, 2021*). However, these studies were conducted on an experimental group of 12 students, making it difficult to generalize the findings to the entire population of this age group. In our age group, children did not use backpacks in any other way than carrying them on their backs, and no child used a backpack with a frame enabling alternative carrying methods— asymmetric behaviors or carrying a school backpack in hand. Research on backpack weight in combination with BMI and the consequences of overloading in children shows significant discrepancies, ranging from studies finding a relationship (*Shultz et al., 2009*) to those denying any correlation (*Pau, Kim & Nussbaum, 2012*). The ones marked as last are differ from ours.

## Normative values of school backpacks

The data regarding recommendations for the weight of a school backpack vary between 5–20% of the child's body weight (*Marinov, Kostianev & Turnovska, 2002*; *Pau, Kim & Nussbaum, 2012*; *Dockrell, Simms & Blake, 2013*; *Malinowska-Borowska & Flajszok, 2020*). Analyzing the backpack's weight in relation to the child's body weight, one might overlook this issue without much concern. If we examined the results within a single group, possibly dividing them by gender, we would not observe any spectacular findings. And this could be considered as correctness in conducting research, as the American Academy of Pediatrics (AAP) (*Moore, White & Moore, 2007*), the American Physical Therapy Association (APTA) (*American Physical Therapy Association, APTA, 2024*), and in Poland, the *Chief Sanitary Inspectorate (GIS) of Poland (2024)* recommend that the weight of a school backpack should be 10–15% of the child's body weight, without distinguishing between children with normal and abnormal body weight. However, when categorizing by body weight, a crucial result emerged: the inability to meet the recommended standards for school backpack weight in the underweight group (as much as 63%) of children attending the same class (with

packed school backpacks with the same textbooks and supplies). Among overweight and obese children, a low percentage of overloaded backpacks was recorded (only 10% in overweight cases and 0.09% in obesity cases). However, this was not due to better control of the backpack's contents but rather a mathematical calculation of proportions. It should be emphasized that the backpack's mass, measured in kilograms, did not differ between the studied groups, which strengthens our considerations. In this group, considering backpacks as compliant with norms—while the same backpack weight would be classified as overloaded for a child with a normal body weight—along with the obtained stabilometric results, dangerously blurs the role of the backpack as a factor worsening postural stability.

Children with normal weight experienced overloaded backpacks three times as often. The backpack poses a daily risk of deteriorating postural stability; therefore, this group of children and their parents should receive special education on the consequences of an overloaded school backpack.

## Postural stability in children with abnormal body mass

Of course, the literature includes studies on school backpacks globally, such as the distribution of weight and its impact on balance (*Negrini & Carabalona, 2002*; *Bataaweel & Ibrahim, 2020*), or the relationship between stability and body posture specifically in groups of girls (*Bieniek & Wilczyński, 2019*).

In our study, interestingly, stabilographic parameters in the group of underweight children did not differ significantly when wearing a school backpack. This likely results from the fact that their musculoskeletal system is not regularly subjected to excessive loads, making it better prepared to bear additional weight. This finding underscores the need for further, detailed studies on this specific group.

Obesity is a real threat that will have health consequences, including, among others, postural stability issues, overload of the musculoskeletal system (*Negrini & Carabalona, 2002*; *Lindstrom-Hazel, 2009*; *Hernández et al., 2020*), exacerbated by the additional burden of school backpack weight on a child's back.

This supports our hypothesis about the deterioration of stabilometric results when adding a load to the back in children with overweight and obesity, despite the fact that, according to recommendations, the backpacks of these children are considered normative.

In our research, we focused on evaluating stabilometric parameters, which are extremely important for maintaining postural stability (*Colné et al., 2008*). Statistics on the prevalence of postural defects in children with overweight and obesity (*Brzęk et al., 2016*; *Maciałczyk-Paprocka et al., 2017*) allowed, and still allow, the exploration of risk factors for the development of these defects, making this topic particularly interesting for the authors from a research perspective.

Our analyzed groups were homogeneous in terms of age as well as stabilometric parameters in the test without a backpack, conducted with eyes open and closed. This served as our starting point and later confirmed the impact of the weight of a school backpack on stabilometric parameters. Statistically significant differences between the groups were noted in the test with a backpack and eyes open, particularly in the Body Weight Distribution, sway path length (SPL), the width of the ellipse, and the height of the

ellipse. The worst results, both in tests with eyes open and closed, were observed in children with obesity. Overweight children improved their symmetry index when carrying a school backpack during the eyes-open test, possibly indicating stimulation of the proprioceptive system by the additional weight, which was not observed in the other groups. The SPL in underweight children was comparable to that of children with a normal body weight. Children with obesity warrant further consideration and research, as this group exhibited the most significant disturbances. The widest and tallest ellipse was observed in children with obesity, indicating that their posture while carrying a school backpack is unstable and highly unsteady, with the center of gravity path being the longest. Wearing the backpack worsened stabilographic outcomes the most in this group compared to others. These findings confirm a significant impairment in proprioception in this group, which is one of the primary sensory systems involved in postural control (*Hue et al., 2007*).

Our research aligns with the findings of French researchers (*Colné et al., 2008*), who indicated that obesity could be a factor contributing to reduced efficiency in performing motor tasks in bipedal postures. Our results also emphasize the importance of postural stability in maintaining the projection of the body's center of mass (COM) within the base of support (*Ivanenko & Gurfinkel, 2018*). To stabilize body segments against gravity and maintain balance, adaptive neuromuscular mechanisms related to COM oscillations are essential (*Gagey et al., 1998*).

This mechanism may explain the deterioration in stabilographic parameters observed after putting on a backpack in a group of overweight and obese children, in whom these mechanisms are impaired. This calls for further research and the implementation of exercises specifically aimed at improving postural stability in children with abnormal body weight. It is important to emphasize that despite the inclusion criterion of correct body posture—any abnormalities excluded a child from the study—it is alarming that stabilographic results in overweight and obese children are so abnormal. The authors have already initiated research on a group of overweight and obese children with diagnosed postural defects.

This is an innovative study that fills a gap and addresses the need for changes in these guidelines. Our research is ongoing, focusing on exploring load limits for children with obesity. Nonetheless, the assumption made by Malaysian researchers in 2017 (*Adeyemi, Rohani & Abdul Rani, 2017*)—that the load value for children with obesity should be 10%—is valid but may be insufficient. In our study, the backpacks of these children ranged from significantly lower values of 3.50% to 20.95% (Mean, 10.20), with nearly half (48.48%) carrying backpacks below the recommended 10% threshold. Despite this, the participants exhibited the poorest stabilometric parameters, suggesting that even the smallest load triggers negative compensatory mechanisms related to postural stability, as evidenced by changes in stabilographic recordings.

The results indicating that the weight of the backpack and the percentage of this weight relative to the child's body mass do not serve as predictors influencing stabilographic parameters seem very intriguing. However, additional loading on the child's back worsens their performance in this area. In underweight children, there are correlations between the backpack's weight, the percentage of this weight relative to the child's body mass, and

stabilographic parameters—relationships that were not observed in groups with increased body mass. In these groups, there was a statistically significant deterioration in results when the child's back was loaded with additional weight. This allows the authors to explore further causes of this phenomenon but also to conclude that, particularly in this group, any additional weight placed on the child's back will affect postural stability. Therefore, it is worth considering these findings and conducting more in-depth research in the group with obesity-related conditions.

In other independent studies conducted over many years, observing the same children over time, a significant systemic problem has been identified (*Brzęk & Plinta, 2016*). Training sessions and workshops carried out as part of projects funded by the Polish Ministry clearly highlight an issue raised by parents: children's class schedules do not allow for lighter packing of backpacks, despite parents' considerable efforts and monitoring of backpack weights. This problem requires a systemic solution (*Brzęk et al., 2017*; *Brzęk et al., 2024*; *Wysocka et al., 2024*). Such a solution will soon be presented in upcoming publications, which are currently at a very advanced stage, focusing on the design of an innovative school backpack, and the results are already promising.

Our research provides hard evidence supporting the necessity of decision-making regarding the weight of school backpacks and systemic solutions. We designed and conducted the study to meet PEDro scale items (*Cashin & McAuley, 2020*), ensuring it was as reliable and valid as possible.

Due to the lack of clear information on what constitutes a safe school backpack weight for children, the varying ranges of back load limits in scientific literature, and the absence of research specifically addressing children with obesity, it is currently not possible to determine definitive weight recommendations. Our study is the first to address this issue in a group of children with abnormal body weight and supports the need to establish normative values for these children.

In our ongoing research, we are focusing on identifying appropriate normative values for these specific groups. At this stage, it is crucial to ensure proper backpack ergonomics, maintain strap symmetry, and reduce backpack weight by following recommendations such as leaving books at school or using a second set of textbooks.

It is difficult to determine the long-term effects of carrying overloaded backpacks due to the lack of research in this area. Indirectly, however, one can indicate the future consequences of prolonged periods of low back pain (LBP) or pain in other body parts that arise during childhood as a result of overloaded backpacks (*Adeyemi, Rohani & Rani, 2015*; *Dockrell, Blake & Simms, 2015*; *Hestbaek et al., 2006*; *Mwaka et al., 2014*), changes in spinal alignment (*Chow et al., 2005*), or chronic asymmetry caused by improper carrying of school backpacks (*Dai et al., 2025*). Our findings regarding postural stability disturbances reinforce the need to seek solutions in this area, as these consequences—stemming only indirectly from backpack overload—may, in adulthood, lead to serious outcomes, affecting health not only on an individual level but also in social and economic dimensions (*Pocovi et al., 2024*). Among the symptoms observed in children with obesity, an increased susceptibility of bones to overload and fractures is noted, as well as a higher risk of osteoporosis in the future. This is a result of increased body weight and the strain on bone tissue. Whether

carrying a backpack further exacerbates this phenomenon remains uncertain and requires further research. However, there is a lack of research based on solid evidence in this regard. This is likely due to numerous confounding factors between developmental age and adulthood.

Due to the fact that the number of children with overweight and obesity is steadily increasing (*Simmonds et al., 2016*), raising concerns among school ergonomics specialists and other professionals, as well as posing challenges for healthcare and public health (*World Obesity Federation & World Obesity Atlas , 2022*). For this reason, the topic of school backpacks cannot be treated as a one-time study. The problem will remain constant for school education, which is why a solution needs to be found now. Statistics on obesity and the increasing issues with postural stability observed in younger school-age children were the impetus for initiating this research. It is essential to carefully consider whether the recommended backpack-to-body weight ratios for overweight and obese children should be the same as those for children with normative body weight. This conclusion is particularly significant in the context of the ongoing obesity epidemic among children and adolescents.

However, this study has certain limitations that should be considered. First, although the sample is representative of the school population, the results cannot yet be generalized to the national level but only to the regional scale, as the research was conducted in various schools and cities in the Silesian region. While the sample size may appear small, it was deemed sufficient based on power analysis. It is important to note that among children with overweight and obesity, it is challenging to meet the criterion of an absence of postural abnormalities, which was crucial to this study. Any postural defects or spinal deformities activate compensatory and adaptive mechanisms that influence posturographic parameters.

A weaker aspect of the study is also the fact that we did not conduct a detailed analysis of the material in terms of physical activity. The children in our study were active at a moderate level, which meant participating in physical education classes according to the schedule. Therefore, the exclusion criteria also included all diseases that could exempt a child from PE classes. However, we did not analyze this issue in details.

In this study, the authors prioritized conducting research while excluding factors that could disrupt postural stability. One could argue that a certain limitation is the lack of research conducted in dynamic conditions; however, this was the premise of this project—to carry out the study in static conditions. It would be worth considering continuing this research in dynamic conditions as well, but in the authors' opinion, it should be conducted in a school environment rather than in laboratory conditions.

## CONCLUSIONS

Recommendations regarding the weight of school backpacks in relation to the body mass of children with increased body weight (overweight and obesity) worsen stabilographic parameters, affecting their postural stability. Alternative percentage load values should be considered, or systemic solutions should be developed for children with overweight and obesity.

## ACKNOWLEDGEMENTS

We would like to thank all children and their families who participated in this study. We would like to thank all the school principals in the Silesia region for their tremendous help in preparing the research sites.

### Funding

The authors received no external funding for this work. The research was funded as part of statutory research of the Medical University of Sile-sia in Katowice under the number: BNW-1-091/N/3/Z. The funders had no role in study design, data collection and analysis, decision to publish, or preparation of the manuscript.

### Grant Disclosures

The following grant information was disclosed by the authors:
Medical University of Sile-sia in Katowice: BNW-1-091/N/3/Z.

### Competing Interests

The authors declare there are no competing interests.

### Author Contributions

- Anna Brzęk conceived and designed the experiments, performed the experiments, analyzed the data, prepared figures and/or tables, authored or reviewed drafts of the article, project administration, and approved the final draft.
- Regina Wysocka-Bochenek performed the experiments, analyzed the data, authored or reviewed drafts of the article, and approved the final draft.
- Jacek Sołtys performed the experiments, analyzed the data, prepared figures and/or tables, and approved the final draft.

### Human Ethics

The following information was supplied relating to ethical approvals (i.e., approving body and any reference numbers):

The Medical University of Silesia in Katowice granted Bioethical approval to carry out the study (The Bioethics Committee of Medical University of Silesia in Katowice, Poland) (KNW/022/KB1/100/I/16/18 and BNW/NWN/0052/KB1/124/II/22/23/24).

### Data Availability

The raw data is available in the Supplementary File.

### Supplemental Information

Supplemental information for this article can be found online at http://dx.doi.org/10.7717/peerj.19353#supplemental-information.

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
