# Peer review of "Children with abnormal body weight dealing with the load of school backpack—is there a need to modify WHO recommendations? A cross sectional study"

_PeerJ, doi:10.7717/peerj.19353_

## Round 0.1 · original submission · Minor Revisions

Dear co-authors,

Based on the reviewers’ comments, I have decided that the manuscript titled: "Children with abnormal body weight dealing with the load of schoolbags - is there a need to modify WHO recommendations? A cross sectional study" requires minor revisions before publication.

The reviewers acknowledge the study’s relevance, methodological rigor, and contribution to the field of biomechanics and public health. However, they highlight the need for greater clarity and detail in certain aspects of the study design, data presentation, and discussion.

Key revisions include providing additional information on participant selection (age, school type, region), specifying experimental conditions (stabilometer environment, schoolbag weight measurement frequency), clarifying statistical tables (Tables 1-4), and ensuring consistency in terminology (schoolbags vs. school backpacks).

Furthermore, some methodological aspects, such as the inclusion of children with pre-existing health conditions and the reliability of the parental questionnaire, should be explicitly addressed.

The discussion should also focus more on the study’s findings rather than resembling an introduction, and any statements regarding study limitations should be moved to the appropriate section.

Lastly, the possibility of integrating findings into recommendations for permissible backpack loads in overweight and obese children should be further explored, with references to additional literature where relevant. Once these minor revisions are incorporated, the manuscript should be well-positioned for publication.

Best regards,

Dr. Manuel Jiménez

Reviewer 1 ·

Basic reporting

no comment

Experimental design

1. Line 99-100, We would like to provide additional information about 1600 younger school-aged youngsters who were chosen at random (such as Communication channels, area throughout the country?, school type private or public, or both et al.)
2. Line 105-106, Did you only focus on the students who carried the bags on both sides? Please be more specific.
3. Line 107 “questionnaire by the parents” What is the questionnaire about? Have the reliability and validity of the tools been assessed? If so, how?"
4. Still Study participant in line 106-111 did the authors include children with heart and circulatory diseases, respiratory system disorders? Please specify
5. Line 136, (n; 33) Is this Ic- obese children? And identified II
6. Line 150, How did you weigh a schoolbag? Every day Monday-Friday or others? How often did you repeat weighing a schoolbag? Please provide more details.
7. Line 157-158, Two trials were carried out: one without a backpack and one with a backpack symmetrically worn on both of the subject's shoulders. How much weight do the kids' schoolbags weigh?
8. It is better to provide an example of the image of the posturographic test with open and closed eyes

Validity of the findings

1. Table 1, Sex, n (%): Please provide the n(%) statistics for both boys and girls. The data in the table for boys/girls is presented as n(%) or X ± SD, which is confusing. Please correct this.
2. df, what are the abbreviations, it is better to present them under table 1,2.
3. Table 2 presents the differences between Ib and Ic; , the difference between Ia and II; , the difference between Ia and Ib;, the difference between Ib and II; and the difference between Ic and II in % SW vs. BW, excluding Ia and Ic. Please review this again."
4. Line 228-231, “This supports our hypothesis about the deterioration of stabilometric results when adding a load to the back in children with overweight and obesity, despite the fact that, according to recommendations, the backpacks of these children are considered normative.” this statement should be in the discussion section.
5. Line 272-307, The first part of the discussion should be concise and not resemble an introduction. Focus on presenting the results of your study and offering a critical analysis.
6. Line 359 COP, what are the abbreviations, it is better to present them in line 359
7. In the discussion section, the content should be organized into two sub-sections based on the purpose. Avoid starting new paragraphs on separate lines, even if they address the same topic or study being discussed.
8. On another objective issue in the discussion part, "the study investigated how stabilographic parameters change under the influence of an external load, such as a school backpack, and whether there is a need to revisit recommendations regarding the permissible backpack load for children with abnormal body mass." Can the results of your study support recommending a specific limit for obese individuals for this purpose? You should include more related papers and theories that suggest an appropriate weight for obese people.
9. There are several sentences in the discussion that focus on limiting your study. Please move all of them to the limitations section.

Additional comments

1. Please write consistently using either 'schoolbags' or 'school backpack
2. Decimals should be used consistently throughout the paper.

Reviewer 2 ·

Basic reporting

a. The article is written in correct academic language. The entire text is understandable and clear.
b. The literature review is extensive and refers to numerous previous studies on the effects of backpack loading on children's musculoskeletal systems. The article aptly addresses WHO standards and health regulations but is somewhat lacking in its discussion of research on the potential long-term effects of backpack loading. The topic of the paper is incredibly important and it is essential to emphasize the long-term effects.
c. The structure of the article follows the journal's guidelines. Table 3 and 4 are very unreadable. Raw data are available and clear.
d. The article is coherent and self-contained, and the results are closely related to the hypotheses. The authors clearly stated the purpose of the study - to test whether backpack loading in different BMI groups affects children's postural stability, and whether current WHO standards are appropriate for all children, regardless of their weight.

Experimental design

a. The hypothesis and objective of the study set by the authors are highly relevant and cover contemporary issues related to the increase in weight problems, children's physical fitness and the increased number of postural defects. In addition, the Hypothesis and objective are in accordance with the journal's guidelines.
b. The authors have clearly formulated an objective and hypotheses that fit into the existing gap in this area.
c. The study was approved by the local bioethics committee. There is no information on under what conditions the stabilometer measurements were carried out. If the test was conducted at school then there is a high risk that many factors could have distorted the test results. Among them: noise, or visual distractions.
d. The stabilometric parameters measured are only listed without explaining what the purpose of their measurement and specific meaning is. To make the research more accessible to people who are not specialists in biomechanics (e.g., teachers, trainers or physiotherapists)

Validity of the findings

a. The data presented is transparent. The statistical tests used are adequate for the analysis performed. The results are presented fairly and completely.
b. A significant limitation of the study is the absence of an evaluation of children's physical activity levels. It is plausible that children with superior physical fitness, exemplified by more developed postural stabilizing muscles, might have attained more favorable outcomes. This factor has the potential to substantially influence the observed results. It is recommended that the authors consider incorporating an assessment of physical activity and physical fitness in subsequent studies to enhance the rigor and relevance of their research.

Additional comments

a. The article addresses an important biomechanical and health issue concerning the effect of backpack weight on the postural stability of children with different body weights. The study fills a gap in the literature, as few previous works have analyzed stabilometry in the context of different BMI categories. The paper was prepared with integrity, and the research methods are well described and justified.
b. Strengths of the article:
i. Relevance of the topic - the study addresses a real-world health problem that may influence future recommendations for backpack loading standards.
ii. Well-designed methodology - the use of the Zebris PDM platform and objective stabilographic measurements ensures high quality data.
iii. Correct statistical analysis - appropriate tests were used to compare groups and assess predictors.
iv. Clearly presented results - the article is logically laid out and the data are well documented in tables and graphs.
c. Areas for improvement:
i. Possible confounders - lack of information on children's physical activity level and fitness, which could affect postural stability and load compensation ability.
ii. Readability of tables, especially 3 and 4.
iii. Detailed description and purpose of stabilometric parameters.
iv. Description of the conditions in which the measurement was performed (a quiet laboratory room or a school with the presence of other children)
v. Possibility of extending the study - postural analysis in motion (e.g., while walking) could provide a more complete picture of the impact of backpacks on children's biomechanics (this was demonstrated in the limitations of the study by the authors).

---

## Round 0.2 · accepted · Accept

Congratulations, your manuscript titled: "Children with abnormal body weight dealing with the load of school backpack—is there a need to modify WHO recommendations? A cross-sectional study" has been considered for publication in PeerJ Journals.

Dr. Manuel Jimenez

Reviewer 1 ·

Basic reporting

Clear

Experimental design

Methods are described with sufficient detail. The revised version is straightforward.

Validity of the findings

Authors have provided additional information about the revisions. It was appropriate and clear.

Additional comments

N/A

Reviewer 2 ·

Basic reporting

No comment

Experimental design

No comment

Validity of the findings

No comment

Additional comments

Everything has been corrected